# Impact of Endocrine Therapy for Cancer on Periodontal Health: A Systematic Review

**DOI:** 10.3390/cancers17183066

**Published:** 2025-09-19

**Authors:** Federica Romano, Francesco Franco, Barbara Mognetti, Giovanni Nicolao Berta

**Affiliations:** 1Department of Surgical Sciences, C.I.R. Dental School, University of Turin, Via Nizza 230, 10126 Turin, Italy; 2Department of Clinical and Biological Sciences, Section of Translational Pharmacology, University of Turin, Regione Gonzole 10, 10043 Orbassano, Italy; francesco.franco@unito.it; 3Department of Life Sciences and Systems Biology, University of Turin, Via Accademia Albertina 13, 10123 Turin, Italy; barbara.mognetti@unito.it

**Keywords:** androgen deprivation therapy, antiandrogens, aromatase inhibitors, breast cancer, hormone therapy, oral health, periodontitis, prostate cancer, selective estrogen receptor modulators

## Abstract

This systematic review aims to comprehensively assess the current evidence regarding the effects of sex hormone treatment on periodontal health in cancer patients. Thirteen articles were identified, twelve of which focused on breast cancer therapy and only one on prostate cancer. The majority of the studies investigated the use of aromatase inhibitors. Despite the heterogeneity among the studies in terms of experimental design, tumor stage, and the drug used, the findings suggest a negative impact of endocrine therapy on periodontal tissues. Periodontal health is often a neglected aspect in the management of cancer patients. It is essential that patients are informed about possible periodontal side effects before initiating treatment and are closely monitored throughout its duration.

## 1. Introduction

Cancer remains a major societal, public health, and economic burden in the 21st century, causing approximately 10 million deaths worldwide across both developing and developed countries [1]. In 2022, there were 20 million new cancer cases, with breast, lung, colorectal, and prostate cancers being the most prevalent [2]. In Europe, breast and prostate cancer accounted for approximately 558,000 and 473,000 new cases in 2022 [3], respectively, while in Italy, an estimated 55,300 new breast cancer (BC) cases and 31,400–40,190 new prostate cancer (PC) cases were projected for 2024 [4]. More than 75% of BCs are driven by estrogen and/or progesterone [5]. Since the early 2000s, PC incidence in Europe increased across most countries before stabilizing or declining in many—especially in Northern Europe, France, Switzerland, and Italy—while mortality generally decreased, except in Eastern and Baltic Europe [6]. In contrast, BC incidence has continued to rise, accompanied by a significant decline in age-standardized mortality (−23.1%) from 1990 to 2019, with Western Europe experiencing the largest decreases [7]. Decline in mortality rates over the last decades is due to improvements in diagnosis and advancements in treatment strategies [8]. Currently, more than 500,000 men and 800,000 women in Italy are living with PC and BC, respectively, with a predicted 5-year survival rate exceeding 80% for patients with either type of cancer [9].

Yet, standard treatment options—such as chemotherapy, radiotherapy, and immune/hormone therapies, whether used alone or in combination—are often associated with a broad range of adverse effects. Notably, oral complications affect approximately 30–35% of patients undergoing cancer treatment [10]. The most frequently reported issues include oral pain, xerostomia, and mucositis, which can severely impair nutritional intake, negatively impact quality of life, and ultimately hinder treatment adherence [11,12]. The largest number of observational studies focused on survivors of head and neck cancers or patients undergoing stem cell transplantation [13].

The impact of cancer therapy on periodontal health remains underexplored, even though gingival conditions are a critical aspect of care for thousands of cancer patients. The tooth-supporting structures—including the gingiva, periodontal ligament, and alveolar bone—are hormone-sensitive tissues [14,15,16]. Sex hormones are crucial regulators of the inflammatory responses and bone remodeling, primarily through cytokine modulation and direct effects on osteoblast and osteoclast precursors [17,18,19]. Lower bone mineral density correlates with an elevated risk of periodontitis and tooth loss [20].

Prostate, breast, and endometrial cancers are known to be driven in part by sex steroid hormones such as androgens, estrogens, and progesterone, which play key roles in cellular growth and proliferation [21]. Increasing evidence suggests that laryngeal cancers may also be responsive to sex hormones, particularly estradiol, through membrane-associated receptors [22].

Within the context of BC, clinical strategies targeting the dysregulated hormonal axis have been implemented since 1896 when Dr. George Thomas Beatson discovered that surgical oophorectomy could induce tumor regression [23]. A major advancement in developing less invasive methods to neutralize hormone-driven tumor growth emerged in 1966 with the development of tamoxifen—a potent selective estrogen receptor modulator that blocks estrogen binding and suppresses the expression of estrogen-dependent genes [24]. Tamoxifen, along with its active metabolites 4-hydroxytamoxifen and endoxifen [25], bind to the estrogen receptor inducing a conformational change that impairs DNA binding and the downstream signaling, thereby reducing the expression of pro-tumorigenic factors such as cyclin D1, c-myc, and bcl-2 [26,27].

In post-menopausal women, aromatase inhibitors (AIs) became a highly effective therapeutic option by inhibiting the conversion of androgens to estrogens, which occurs predominantly in extra-gonadal tissues, such as the adrenal glands, skin, muscles, adipose tissue and even within the tumor microenvironment [28,29].

In contrast pre-menopausal women primarily produce estrogen via ovaries, thus AI monotherapy may trigger compensatory gonadotropin release, stimulating further ovarian estrogen production. To mitigate this, medical gonadal ablation is employed using gonadotropin-releasing hormone agonists, luteinizing hormone–releasing hormone agonists, or antagonists, which suppress ovarian function and enhance the efficacy of AIs in this population [30,31].

Notably, similar hormone suppression strategies are, also, first-line pharmacological treatment for advanced PC in men, which is likewise driven by androgen signaling [30,31].

As a result, hormonal therapy—targeting these hormone-dependent pathways—has become a cornerstone in the treatment of these cancers, both in adjuvant and metastatic settings, with the goal of inhibiting cancer cell proliferation [30]. The selection of a specific endocrine regimen is typically guided by a combination of clinical, pathological, and genetic factors, including the patient’s menopausal status [31].

Consequently, long-term survivorship issues, including oral health, have become critical components of cancers’ care and follow-up. In this context, understanding the impact of endocrine therapy on periodontal health is increasingly important. Recent reports have indicated an increased risk of periodontitis in patients undergoing endocrine therapy, with hormone therapy–induced osteoporosis identified as one of the main contributing factors. However, the available data remain debated and often yields inconclusive results [32]. Figure 1 depicts a proposed mechanism linking hormone therapy for BC to the onset of periodontitis.

Further scientific evidence is needed to inform preventive strategies and enhance the planning of healthcare resources. Therefore, the aim of this study was to systematically review the literature on the effects of endocrine therapy on periodontal parameters and tooth loss in cancer patients.

## 2. Materials and Methods

This systematic review is reported according to the PRISMA (Preferred Reporting Items for Systematic Reviews and Meta-analyses) [33] and MOOSE (Meta-analysis of Observational Studies in Epidemiology) guidelines [34]. The protocol for the systematic review of our study was not registered for PROSPERO.

### 2.1. Focused Question

This systematic review was designed to address the following two distinct focused questions (FQs):
(FQ1) In patients with cancer, what is the effect of endocrine therapy on periodontal status and tooth loss compared to nonusers or healthy individuals?(FQ2) Do different sex hormone treatments produce varying effects on periodontal tissues?

### 2.2. Eligibility Criteria

Eligibility criteria for this systematic review were established based on the PICOS framework as follows:

Population (P): Individuals diagnosed with any type of cancer, without restrictions on age, sex, or country of origin.

Intervention (I): Treatment with hormone therapy, with no limitations regarding the specific drug, dosage, or duration.

Comparison (C): Individuals not receiving endocrine therapy, healthy controls or none.

Outcome measures (O): The primary outcome was the periodontal status in terms of prevalence of gingivitis/periodontitis, reported either as a percentage or number of affected individuals, or patient’s self-reported perception of their periodontal health, assessed through questionnaires or visual analog scales (VAS), or mean number of remaining or lost natural teeth. Secondary outcomes were mean clinical attachment level (CAL), defined as the distance between the cementoenamel junction (CEJ) and the bottom of the sulcus/periodontal pocket, probing pocket depth (PPD), defined as the distance between the gingival margin and the bottom of the sulcus/pocket, reported in mm, the percentage/number of sites harboring plaque (plaque index, PI) or displaying inflammation (gingival index, GI, or bleeding on probing, BoP), and the number of decayed and filled teeth.

Types of studies (S): All observational studies, retrospective or prospective, cross-sectional, cohort or case–control, involving ≥10 human subjects. Excluded were case reports, expert opinions, conference abstracts, systematic reviews, in vitro and animal studies. In case of duplicate publications involving the same population or subpopulation, the most recent or most comprehensive report was included.

### 2.3. Literature Search Strategy

Three electronic databases (MEDLINE via PubMed, Scopus, Web of Science) were systematically searched by two independent authors up to June 2025. A comprehensive search strategy was devised in MEDLINE to identify articles including three distinct concept blocks: (i) cancer, (ii) hormone-based antineoplastic treatment and (iii) gingival/periodontal health status. For each concept block, a combination of medical subject headings (MeSH) terms and free-text words were identified and used. The search strategy was then adapted for the other databases (Table 1). The search was restricted to studies published in English-language journals, with no limitations on the date of publication. In addition, both reviewers manually screened the reference lists of all included studies and relevant systematic reviews to identify additional eligible articles. Any study identified by at least one reviewer during this process was advanced to the next stage of review.

### 2.4. Article Selection and Reviewer Agreement

Following the removal of duplicates, the titles and abstracts of retrieved articles were independently screened by two reviewers. Studies that appeared to meet the inclusion criteria or for which eligibility could not be determined from the abstract alone were retrieved in full for further assessment. The reasons for exclusion after full-text review were documented in detail. Disagreements between reviewers were resolved through discussion; if consensus could not be reached, a third author was consulted. Inter-reviewer reliability during the screening process phases was assessed using the kappa correlation coefficient.

### 2.5. Data Extraction and Risk of Bias Assessment

All included studies underwent data extraction carried out independently and in duplicate by two reviewers. Multiple publications based on the same dataset were included if they reported on different subgroups or analytical aspects. The first three studies were used for calibration purposes during the data extraction phase. The following information was extracted from each study using predefined standardized forms: publication year, study design, country and setting, participant characteristics, cancer type, details of endocrine therapy (drug, dosage, and duration), outcome measures of interest, authors’ conclusions, and sources of funding.

Risk of bias assessment and study quality were independently assessed by two authors using the Newcastle–Ottawa Scale (NOS) for cohort studies and the Joanna Briggs Institute (JBI) Critical Appraisal Checklist for analytical cross-sectional studies. The NOS evaluates participant selection, comparability of study groups, and assessment of outcomes or exposures, with a maximum score of 9 stars [35]. Studies scoring 7 or more were classified as having a low risk of bias. The JBI checklist assessed criteria such as sample selection, measurement of exposure and outcome, and control for confounding variables. Each item was rated as ‘yes’, ‘no’, ‘unclear’, or ‘not applicable’, with overall study quality judged based on the number of criteria met [36]. Any discrepancies in scoring were resolved through discussion.

### 2.6. Summary Measures and Synthesis of Results

The included studies were first narratively summarized on their key characteristics, categorized by cancer type and antineoplastic treatment. A meta-analysis was conducted when at least two studies with comparable designs were available. Data on periodontal status, in terms of prevalence of periodontitis and mean difference in CAL between case and control groups, were combined in the meta-analysis and presented as weighted means and 95% confidence intervals (CI). The association between periodontitis and BC treatment was assessed by calculating pooled odds ratio (OR) with corresponding 95% CI.

Statistical heterogeneity among studies was evaluated using Cochran’s Q-test (*p* < 0.05 considered significant) and I^2^ statistic, with thresholds of 25%, 50%, and 75% indicating low, moderate, and high heterogeneity. When significant heterogeneity was detected, a random-effect model was applied using the generic inverse variance method [37].

Forest plots were generated to visually depict the effect sizes of individual studies and the overall pooled estimate. All statistical analyses were conducted using the OpenMeta (Analyst) software (version v. 024.0). A *p* value < 0.05 was considered statistically significant.

## 3. Results

### 3.1. Study Selection

Figure 2 shows the PRISMA workflow for study identification, screening and inclusion. A total of 422 publications were identified through electronic database search after the removal of duplicate records. No additional citations were identified through manual searching. Of these, 31 studies were excluded due to language restriction, and 367 were excluded following title and abstract screening as they did not align with the PICOS criteria (kappa = 0.92, 95% CI: 0.85–0.97). After full-paper assessment, 11 articles were removed (reasons for exclusion reported in Figure 2), while 13 articles [38,39,40,41,42,43,44,45,46,47,48,49,50] were finally included into the qualitative analysis and considered for data extraction (kappa = 0.89, 95% CI: 0.81–0.96).

### 3.2. Study Characteristics

The selected literature, published from 2007 to 2025, comprised 10 cross-sectional studies, 1 cohort study, and 2 prospective observational studies. The articles by Eagle et al. [41] and Taichman et al. [40] appear to be consecutive publications derived from the same cohort described in Taichman et al. [42], presenting findings over an 18-month follow-up period. Although based on the same population, each publication focused on distinct outcomes.

All studies, except for two that utilized nationwide health datasets, were conducted in hospital settings or university clinics across six countries, with the United States being the most represented (*n* = 6). Most focused on the hormone therapy in post-menopausal women with BC [38,39,40,41,42,43,44,45,46,47,48,49], with only one assessing the effects of androgen deprivation therapy (ADT) in nonmetastatic PC [50]. Among the BC-related studies, five [38,39,40,41,42] investigated the impact of aromatase inhibitors (AIs), including anastrozole, exemestane and letrozole, on periodontal health, two studies [43,44] focused on tamoxifen, while the remaining five assessed the effects of both tamoxifen and AIs [45,46,47,48,49]; of these three provided separate results for each drug.

Periodontitis was defined using various criteria, including the Periodontal Screening and Recording Index [51], the 2017 World Workshop classification [52], clinical measures (typically clinical attachment loss ≥ 3 mm), or subjective assessments based on questionnaires. In some cases, however, non-specified definitions of periodontal diseases were used. Study characteristics are summarized in Table 2.

### 3.3. Synthesis of the Results

The data retrieved from the included studies to address FQ1 are presented below, categorized by cancer type and pharmacological intervention.

#### 3.3.1. Impact of Hormone Therapy on Periodontal Health in Breast Cancer

Aromatase inhibitors therapy

In a cross-sectional study conducted by Taichman et al. [42], periodontal health was significantly compromised in 29 postmenopausal women with early-stage BC who had recently initiated adjuvant AI therapy (anastrozole, exemestane, or letrozole; mean treatment duration of 5.7 ± 3.1 months), compared to 29 healthy controls not undergoing AI therapy. Women receiving AIs exhibited significantly higher prevalence of severe periodontitis (31.0% vs. 6.9%; *p* = 0.03), poorer overall oral hygiene (55.4 vs. 16.3 plaque sites; *p* = 0.03), greater number of gingival bleeding sites (27.8 vs. 16.7; *p* < 0.02), and higher mean worst-site attachment loss (5.2 mm vs. 4.0 mm; *p* < 0.01) than controls. However, no statistically significant differences were observed between the two groups in terms of mean PPD and number of teeth.

Two follow-up prospective studies [40,41], involving the same cohorts, further confirmed the progression of periodontal deterioration among the AI-treated BC group over an 18-month period. These women experienced a significant worsening in all clinical periodontal parameters and more extensive radiographic alveolar bone loss, compared to the healthy control group (all *p* < 0.05). Calcium supplementation appeared to offer a protective effect against alveolar bone height loss in the AI group (*p* = 0.005). Furthermore, women undergoing AI therapy reported a more negative perception of their oral and periodontal health than their healthy counterparts.

The study by Ferrillo et al. [39] investigated periodontal health in post-menopausal BC women with vitamin D deficiency who were undergoing AIs therapy. Among the participants, approximately 56% had osteoporosis, 37% had osteopenia, and 29% were smokers. The results showed that only a minority of patients exhibited periodontal health and effective plaque control at home. Notably, 65.8% were diagnosed with periodontitis, and around 47% had a full-mouth plaque score exceeding 50%. Using a regression-based machine learning model, the authors found strong correlations between the Decayed, Missing, and Filled Teeth (DMFT) score and some clinical variables, including the daily use of interdental cleaning devices, smoking habits, and serum vitamin D levels.

In contrast to previous findings, the study by de Souza et al. [38] reported no significant differences in periodontal status between BC women using AI and healthy controls. The overall prevalence of periodontal diseases was comparable between the two groups, although the proportion of women with Stage II-IV periodontitis was slightly higher among AI users (72.5% vs. 67.9%). Also, the odds of reporting a higher oral health–related quality of life were 1.18 times greater among women receiving AIs relative to controls. However, 47.5% of AI-treated women were receiving antiresorptive drugs and 22.5% were on denosumab, compared to 31.0% and 3.5%, respectively, in the control group (*p* = 0.013 and *p* = 0.035, respectively).

Tamoxifen therapy

de Araujo-Sansever et al. [44] investigated the association between tooth loss (TL) and duration of tamoxifen therapy in 140 BC women (81.3% with ductal carcinoma and 84.1% with stage I-II) and found, among participants who used tamoxifen for over one year, a greater average number of missing teeth (13.99 vs. 10.45; *p* = 0.030). In an age-, education-, dental care-, xerostomia-, and caries-adjusted logistic regression, the likelihood of having more than 12 missing teeth was 2.75-fold higher among women treated with tamoxifen for more than 1 year (95% CI: 1.06–7.12).

Consistently, Julca-Baltazar et al. [43] observed greater TL in women had been taking tamoxifen for more than one year compared to BC women who had not used it (*p* = 0.025), as well as in older women compared to younger ones (*p* = 0.030). However, reduced TL was found in patients who used tamoxifen—regardless of treatment duration– if they had also received any form of systemic therapy, except in the subgroup that received both tamoxifen for more than one year and radiotherapy, where TL remained comparatively high (*p =* 0.055).

Endocrine therapy (Aromatase inhibitors or Tamoxifen)

In the study by de Sire et al. [48], poor periodontal and dental health were observed in 122 post-menopausal women with invasive BC undergoing tamoxifen or AI adjuvant hormone therapy. Most participants were diagnosed with moderate (63.1%) or severe (15.6%) periodontitis and exhibited poor plaque control (51.6%), with over half of the examined sites showing plaque accumulation. The average DMFT index was high at 16.07 ± 7.05. Although all women were assessed prior to starting anti-osteoporotic treatment, no significant correlations were found between oral health parameters and bone mineral density or vitamin D status.

Supporting these findings, Park et al. [46], using data from the NAHNES, reported that a higher proportion of adult BC survivors on hormone therapy (AI or tamoxifen) had decayed teeth, gum problems, and were advised to seek immediate dental care, compared to those not undergoing such therapy.

#### 3.3.2. Impact of Hormone Therapy on Periodontal Health in Prostate Cancer

In the only study identified on PC patients [50], the prevalence of periodontitis was 80.5% among adult and elderly men undergoing ADT for an average of 1.5 years, compared to just 3.7% in PC patients not receiving ADT (OR 3.33, 95% CI: 1.07–10.35). Men undergoing ADT treatment also exhibited significantly deeper probing depths and higher bleeding scores than controls (both *p* < 0.001). Although tooth mobility was higher in the ADT group, the difference did not reach statistical significance (*p* = 0.074). Notably, the specific class of ADT drugs used was not reported.

#### 3.3.3. Comparison Between Different Hormone Therapies (FQ2)

Only three studies stratified clinical data based on the use of tamoxifen or AIs for BC [45,47,49]. In the study by Ustaoğlu et al. [47], tamoxifen use was not associated with adverse oral health outcomes, whereas AI therapy was linked to significantly greater number of missing teeth (*p* < 0.001) and more severe clinical attachment loss (*p* = 0.042). However, the AI group was older than the other two groups. No significant differences in periodontal parameters were found between different AI medications. Nonetheless, among AI users, those on therapy for more than five years exhibited poorer plaque control and fewer remaining teeth compared to those with shorter treatment durations.

In another study, Taichman et al. [49] assessed postmenopausal women with early-stage BC who had received chemotherapy, tamoxifen, or AIs for at least three months, focusing on their perceptions of oral and dental status. Interestingly, the control group reported worse perception of both their dental (mean score: 2.73 vs. 3.47; *p* < 0.001) and gingival health (2.93 vs. 3.42; *p* < 0.05) compared to the combined breast cancer groups. When comparing AI and tamoxifen users, women on tamoxifen reported a better perception of their dental health. Additionally, patients undergoing chemotherapy were more likely to be aware that cancer treatment could impact oral health compared to those receiving tamoxifen or AIs (93% vs. 55% and 56%, respectively; *p* < 0.001).

Conversely, in the only retrospective cohort study included in this systematic review [45], hormone therapy demonstrated a protective effect against periodontitis in women with BC, based on data from Taiwan’s National Health Insurance Research Database. After a 12-year follow-up, women who received tamoxifen or AI therapy had a significant lower risk of developing periodontitis compared to those who did not receive hormone therapy.

### 3.4. Quantitative Analyses of the Results

Regarding FQ1, pooled data analyses assessing the impact of AI use (anastrozole, exemestane and letrozole) on periodontal health in BC populations are presented in Figure 3, Figure 4 and Figure 5. Meta-analyses were not feasible for studies involving tamoxifen or for FQ2 due to limited data availability or methodological heterogeneity. The fixed-effect meta-analysis (Figure 3) demonstrated a weighted prevalence of moderate to severe periodontitis of 76.0% (95% CI: 70.5 to 81.5) among post-menopausal BC women receiving AIs, with no observed heterogeneity (I^2^ = 0%).

These individuals were 1.55 times more likely to experience periodontitis compared to healthy women not taking AIs (Figure 4); however, this association did not reach statistical significance (*p* = 0.552).

As reported in Figure 5, no significant group differences were also observed across the three studies [3,42,47] evaluating the impact of AI intake on CAL in BC women compared to healthy controls (weighted mean difference: 0.28 mm, 95% CI: −0.204 to 0.764; *p* = 0.256; I^2^: 80.3%).

### 3.5. Risk of Bias

The only cohort study included in the systematic review was found at low risk of bias (Table A1, Appendix B), while all the cross-sectional studies, except for two, were found at moderate-to-high risk of bias (Table A2, Appendix B).

## 4. Discussion

To our knowledge, this is the first systematic review to comprehensively evaluate the impact of endocrine therapy for cancer treatment on periodontal health. Although such therapy is a cornerstone in the management of hormone-sensitive cancers—significantly improving survival and delaying disease progression—its oral side effects are frequently underreported in the literature. In the present review, only twelve studies related to BC and one study on PC were identified. Despite considerable heterogeneity among the included studies in terms of experimental design, tumor stage, and types of drugs used, the overall findings suggest a potentially negative effect of endocrine therapy on periodontal tissues. However, pooled data analysis on AIs revealed no significant difference in periodontitis prevalence and CAL severity among post-menopausal BC women compared to control groups. Due to insufficient data, a meta-analysis could not be performed for tamoxifen use.

Women with hormone-receptor positive BC are typically prescribed adjuvant endocrine therapy for at least five years following surgery [31,53]. This therapy, which includes AIs and tamoxifen, reduces estrogen levels, with varying effects on periodontal tissues depending on the class of drug used [31,53].

AIs are widely administered to postmenopausal women as they fully suppress estrogen production. Aromatase, a cytochrome P450 hemoprotein-containing enzyme, catalyzes the conversion of estrogens to androgens primarily in fat cells, as well as in various peripheral tissues, including breast and muscle tissue [54]. Third-generation AIs (anastrozole, exemestane and letrozole) are commonly recommended as part of the therapeutic regimen for BC due to their superior efficacy in reducing recurrence and their overall tolerability compared to tamoxifen [54]. The majority of studies included in this review indicated that AI therapy was associated with a higher prevalence of severe periodontitis, increased plaque accumulation, a greater proportion of bleeding sites and more pronounced alveolar bone resorption compared to age-matched, healthy controls [39,42,47]. In addition, periodontal conditions were found to worsen over an 18-month observational period among AI users [41], who also reported poorer self-perceived oral and periodontal health compared to healthy controls, highlighting the broader impact of AI treatment on both clinical and patient-reported outcomes [40].

In contrast, de Souza et al. [38] found no significant differences in periodontal status between BC women using AI and healthy controls. Interestingly, they also observed a better oral health-related quality of life in the AI group, aligning with findings by Taichman et al. [49]. These conflicting findings may be, at least in part, attributed to differences in periodontal assessment methods, adjustment for confounding factors such as smoking and oral health practices, differences in the duration of AI therapy, and the concurrent use of adjunctive medications. Epidemiological and clinical studies have consistently demonstrated the detrimental effect of smoking on periodontal health and the critical role of effective home plaque control in the prevention and progression of periodontitis [55,56]. The reduced salivary flow rate observed in endocrine therapy-treated women may predispose to plaque accumulation increasing the risk of gingival inflammation and dental decay [57]. Additionally, anti-estrogen therapies may adversely affect patients’ mental well-being—causing fatigue and depression—which in turn can lead to neglect oral hygiene and general healthcare [58].

Estrogen deficiency is also known to reduce bone mineral density and contribute to alveolar bone loss [18,20]. Supplementation with calcium and vitamin D, along with the administration of antiresorptive drugs, may offer a protective effect against alveolar bone height loss in this patient population [40,59].

Beyond the role of estrogen depletion in disrupting periodontal tissue homeostasis, AIs have also been suggested to exert direct effects on periodontal cells by modulating the inflammatory response. Eagle et al. reported elevated levels of proinflammatory cytokines in the saliva of AI users [41]. In vitro, stimulation of human gingival fibroblasts and endothelial cells with anastrazole led to increased expression of collagen and extracellular matrix proteins, enhanced vessel permeability and upregulation of proinflammatory cytokines in presence of *P. gingivalis* lipopolysaccharide [60].

For premenopausal women tamoxifen—a nonsteroidal triphenylethylene derivative that blocks estrogen receptors—is commonly prescribed [53]. Prolonged tamoxifen treatment exceeding one year has been linked to greater tooth loss [43,44], although it remains unclear whether it was primarily due to periodontal disease or other dental pathologies. Compared to AIs, tamoxifen appears to have a milder impact on periodontal health [47,49]. Indeed, Ustaoğlu et al. [47] observed a greater number of missing teeth and more severe clinical attachment loss in patients receiving AIs compared to those treated with tamoxifen, particularly among those who had been on therapy for more than five years, with no significant differences among specific AI medications. Consistently, women receiving tamoxifen reported a more favorable perception of their dental and gingival health compared to AI users [49].

Tamoxifen’s effect might partially result from blocking beta receptors in acinar and ductal cells of the major and minor salivary glands, thereby altering both the quantity and composition of saliva [61]. In addition, changes in the oral microbiome, predisposing to oral diseases, have been observed in the animal model and have been linked to estrogen deficiency [62].

Only one dated cross-sectional study has evaluated the impact of ADT on periodontal health in men with PC [50]. The prevalence of periodontitis was found to be three times higher in the ADT group compared to PC patients not receiving ADT, even after adjusting for confounding factors such as age, race, smoking and periodontal treatment history. This association persisted despite adequate plaque control and was independent of the duration of ADT.

ADT, a primary standard treatment for PC, reduces serum androgen levels [30]. Androgen receptors have been identified on periodontal tissues, suggesting a direct role of sex male hormones in maintaining periodontal homeostasis [63]. Lower testosterone levels have been linked to increased production of proinflammatory cytokines, including interleukin (IL)-6, tumor necrosis factor (TNF)-alpha and IL-1 beta, by osteoclasts and fibroblasts, promoting bone resorption and connective tissue breakdown [16]. Additionally, ADT impairs host defense mechanism by downregulating the expression of E-selectin and vascular adhesion molecule-1 in endothelial cells, both of which are essential for the trans-endothelial migration of phagocytic cells [64]. Consistently, ADT has been associated with elevated salivary levels of matrix metalloproteinase-8, potentially reflecting impaired polymorphonuclear leukocyte chemotaxis, increased collagenase and elastase activity, and reduced collagen synthesis by fibroblasts, all contributing to periodontal tissue breakdown [65].

### 4.1. Strenght and Limitations

A major strength of the present study lies in its thorough, systematic search for relevant literature across multiple databases, ensuring an exhaustive selection of observational studies assessing the impact of endocrine therapy in cancer patients. The inclusion criteria were deliberately broad, encompassing any hormone-sensitive cancer population to enhance the generalizability of the findings. Study selection and data extraction were independently performed by two reviewers, with a high level of inter-reviewer agreement, which strengthens the reliability of the process.

However, several limitations should be acknowledged. Significant heterogeneity was observed among the included studies in terms of design, cancer stage, definitions of periodontal disease, methods of periodontal parameter assessment, and adjustment for confounding factors. These inconsistencies limit the comparability of results and hinder the ability to draw definitive conclusions. Moreover, the overall quality of evidence was low, with a high risk of bias across studies, further restricting the strength of the findings.

### 4.2. Implications for Research and Clinical Practice

Given that hormone therapy is typically administered for a minimum of 5 years [30], its oral side effects persist over time, significantly impacting the long-term oral health and quality of life of cancer survivors [57]. Poor oral health (bleeding, mobile, or missing teeth) and compromised orofacial functions (chewing, swallowing, and speech) may lead to physical weakness due to altered eating habits [41]. These physical symptoms often coexist with psychosocial issues such as distress, anxiety, and fatigue and, in some cases, social isolation [41]. Such effects not only impair overall well-being, but may also negatively influence treatment adherence, potentially reducing the effectiveness of cancer treatment.

These findings highlight the urgent need for more high-quality, longitudinal studies assessing the effects of endocrine therapy on periodontal health across different cancer populations. Current evidence remains limited, with considerable heterogeneity in study design and quality. Nevertheless, the present study highlights the significant yet often overlooked aspect of oral health in patients undergoing hormonal cancer therapies. It is clear that interdisciplinary collaboration integrating oncologists, dentists, and dental hygienists is essential to ensure that potential side effects of endocrine therapy on oral tissues are not neglected but properly addressed. Another concern is the bidirectional links between periodontal disease and systemic conditions such as diabetes and cardiovascular disease, that should foster even more the evaluation of periodontal status during cancer treatments. Overall, such multidisciplinary approach may not only improve patient quality of life, but also support better adherence to endocrine therapy and enhance overall treatment outcomes.

## 5. Conclusions

Within the limitations of the available evidence, the present findings suggest that endocrine therapy may have a detrimental effect on periodontal tissues. Despite its clinical relevance, periodontal health is often overlooked in the comprehensive care of cancer patients. It is therefore essential that patients are informed of its potential oral side effects prior to treatment initiation and are monitored regularly throughout the course of therapy to enable early detection and management of oral health-related complications.

## Figures and Tables

**Figure 1 cancers-17-03066-f001:**
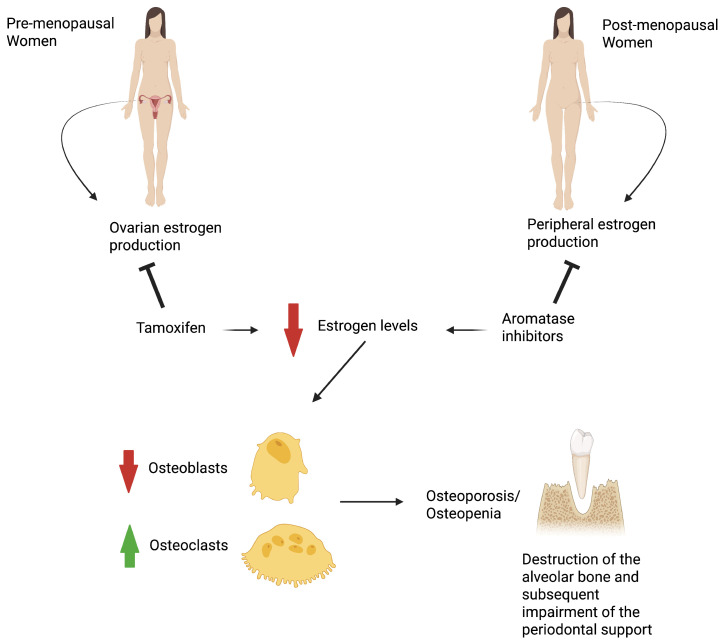
Proposed mechanism linking endocrine therapy in breast cancer patients to the development of periodontitis.

**Figure 2 cancers-17-03066-f002:**
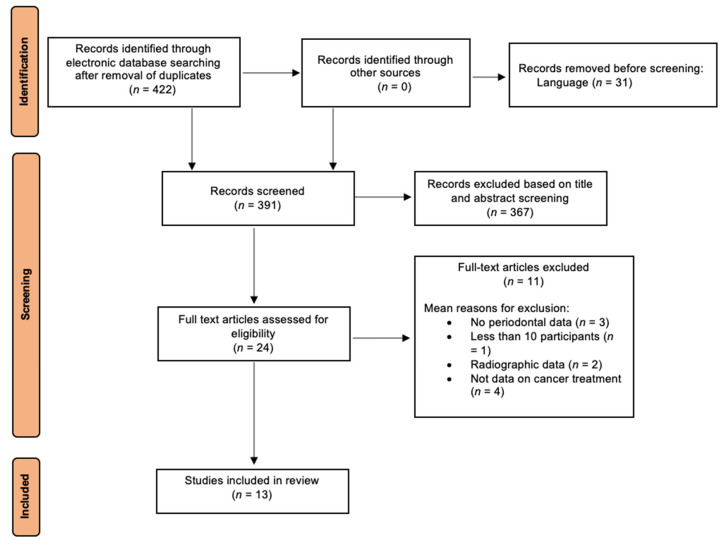
PRISMA flow diagram for screening and identification of publications (for the PRISMA checklist see Appendix A).

**Figure 3 cancers-17-03066-f003:**
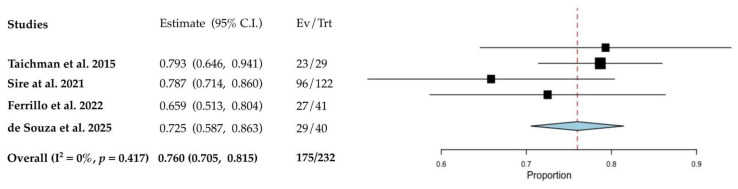
Forest plots from fixed-effect meta-analysis on the prevalence of periodontitis in BC women on AI [38,39,42,48]. The red dashed line intersecting the diamond represents the pooled point estimate, derived from the aggregated results of all studies included in the meta-analysis.

**Figure 4 cancers-17-03066-f004:**
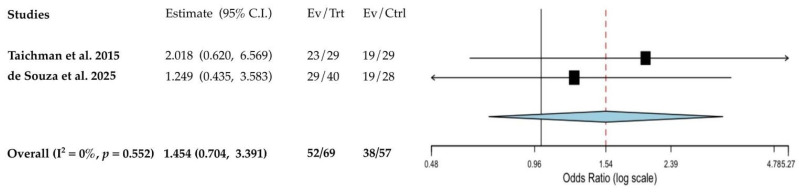
Fixed-effect meta-analysis forest plot assessing the likelihood of periodontitis in breast cancer patients on AI therapy compared to healthy controls (odds ratio, 95% CI) [38,42]. The red dashed line intersecting the diamond represents the pooled point estimate, derived from the aggregated results of all studies included in the meta-analysis.

**Figure 5 cancers-17-03066-f005:**
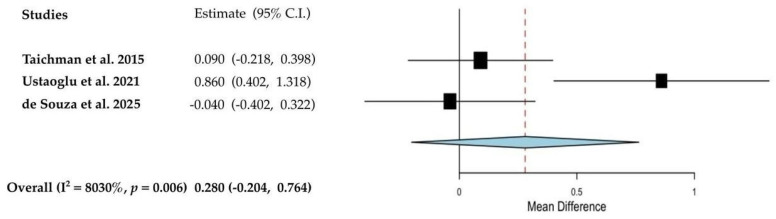
Forest plot from random-effect meta-analysis evaluating the impact of AI intake on clinical attachment level compared to healthy controls [38,42,47]. The red dashed line intersecting the diamond represents the pooled point estimate, derived from the aggregated results of all studies included in the meta-analysis.

**Table 1 cancers-17-03066-t001:** Search Strategy for Medline.

(“neoplasms”[MeSH Terms] OR “adenocarcinoma”[Title/Abstract] OR “carcinoma”[Title/Abstract] OR “cancer”[Title/Abstract]) AND (“antineoplastic agents, hormonal”[MeSH Terms] OR “androgen receptor antagonist*”[Title/Abstract] OR “anti androgen*”[Title/Abstract] OR “LHRH”[Title/Abstract] OR “luteinizing hormone-releasing hormone”[Title/Abstract] OR “luteinizing hormone releasing hormone analog*”[Title/Abstract] OR “luteinizing hormone releasing hormone agonist*”[Title/Abstract] OR “androgen”[Title/Abstract] OR “androgen deprivation therap*”[Title/Abstract] OR “selective estrogen receptor modulator*”[Title/Abstract] OR “endocrine therap*”[Title/Abstract] OR (“aromatase inhibitor*”[Title/Abstract] OR “hormone therap*”[Title/Abstract]) OR “tamoxifen”[Title/Abstract]) AND ((“periodontitis”[MeSH Terms] OR “mouth diseases”[MeSH Terms] OR “oral health”[MeSH Terms] OR “gingivitis”[MeSH Terms] OR “tooth loss”[MeSH Terms] OR “periodont*”[Title/Abstract] OR “gingiv*”[Title/Abstract] OR “oral health”[Title/Abstract] OR “oral condit*”[Title/Abstract]) OR “oral stat*”[Title/Abstract])

**Table 2 cancers-17-03066-t002:** Characteristics of the included studies categorized by cancer type and pharmacological intervention.

**First Author Year, Country**	**Focused Question**	**Study Design, Setting, Time Period, Source of Funding**	**Sample Size (*n*)**	**Test Group (*n*)**	**Control Group (*n*)**	**Definition of Periodontal Disease**
*Studies on breast cancer*						
*Aromatase inhibitors*						
de Souza et al., 2025, Brazil [38]	FQ1	Cross-sectional, Cancer Unit Mater Dei Hospital, October 2019 to August 2021, no external funding	72	Adult women with BC assuming AIs (*n* = 40), of whom 24 taking letrozole, 14 anastrozole, and 2 exemestane, having at least 6 teeth, 40% of them assuming biphosphonates, mean age 61.65 yrs, 67.5% White, 20% Black, 15% mixed, 27.5% smokers	Adult women without BC not using AIs (*n* = 28), having at least 6 teeth, 18.7% of them using bisphosphonates, mean age 60.64 yrs, 75% White, 10.7% Black, 14.3% mixed, 42.9% smokers	2017 World classification of periodontal health and Stage I–IV periodontitis
Ferrillo et al., 2022, Italy [39]	FQ1	Cross-sectional, University Hospital “Maggiore della Carità” Novara, April 2021 to March 2022, no external funding	41	Post-menopausal BC women with vitamin D deficiency undergoing AI therapy, 29.3% smokers, mean age 66.1 ± 8.47 yrs, 29.27% smokers		Periodontal Screening and Recording Index (PSR)
Taichman et al., 2016 and Eagle et al., 2016 USA [40,41]	FQ1	Longitudinal study, University of Michigan Hospital, April 2009 to September 2013, no external funding. 18-month follow-up examination of the study by Taichman et al. 2015 [42]	58	Post-menopausal women (*n* = 29) with BC (Stage I-IIIA with no evidence of metastatic disease) treated with any AI (anastrozole, exemestane, or letrozole), mean age 61.7 ± 7.6 yrs, 89.7% White and 10.3% non-White, 3.4% smokers	Post-menopausal women without BC (*n* = 29), mean age 61.6 ± 5.4 yrs, 89.6% White and 10.4% non-White, 3.4% smokers	Subjective periodontal and dental health based on 5-point scale questions from the NHANES and 11 binary questions on periodontal disease-related symptoms
Taichman et al., 2015, USA [42]	FQ1	Cross-sectional, University of Michigan Hospital, April 2009 to September 2010, no external funding	58	Post-menopausal women (*n* = 29) with early-stage BC (Stage I-IIIA) on AI adjuvant treatment (anastrozole *n* = 20, exemestane *n* = 2, and letrozole *n* = 7, within 2 to 11 months of start), having > 15 teeth, mean age 61.7 ± 7.6 yrs, 89.7% White and 10.3% non-White, 3.4% smokers	Post-menopausal women without BC (*n* = 29), not on AI therapy, having > 15 teeth, mean age 61.6 ± 5.4 yrs, 89.6% White and 10.4% non-White, 3.4% smokers	Case of periodontitis defined as at least one site with AL of ≥3 mm, and classified as mild (AL = 3 mm), moderate (AL ≥ 4 mm but <6 mm) and severe (AL ≥ 6 mm)
*Tamoxifen*						
Julca-Baltazar et al., 2024, Peru [43]	FQ1	Cross-sectional, High Complexity Hospital and Regional Institute of Neoplastic Diseases (Trujillo, Peru), July to September 2023, no external funding	200	Adult women with BC assuming tamoxifen (*n* = 100)	Adult women with BC not assuming tamoxifen (*n* = 100)	NR
de Araujo Sensever et al., 2022, Brazil [44]	FQ1	Cross-sectional, Hospital Southern Brazil, January to August 2017, no external funding	140	Adult women with BC taking tamoxifen for up to 12 months (*n* = 41)	Adult women with BC taking tamoxifen for more than 12 months (*n* = 97)	NR
**Outcome Measures**		**Impact on Periodontal Health**			**Additional Findings**	
*Aromatase Inhibitors*						
PI; BoP; PPD; CAL; DMFT		Cases Periodontitis stage II–IV: 72.5%; mean PPD: 1.89 mm; mean CAL: 2.24 mm; mean BOP: 7.26%; mean PI: 10.95%; mean DMFT: 21.23 Controls Periodontitis stage II–IV: 67.9%; mean PPD: 2.02 mm; mean CAL: 2.28 mm; mean BOP: 8.13%; mean PI: 20.13%, mean DMFT: 20.00			PI of controls was significantly higher than that of women who used AIs. The groups were similar for DMFT index, PPD, CAL, and BoP. However, the number of patients with stage II–IV periodontitis tended to be higher in the case group.	
OHI; GBI; PCR; DMFT; WTCI		Moderate periodontitis: 63.1% Severe periodontitis: 15.6%; OHI > 3: 43.9%; PCR index > 50%: 46.3%; DMFT (mean ± SD): 16.07 ± 7.05; WTCI grade 2: 36.6%			High prevalence of osteoporosis (56.10%) and smokers (29.3%). Prevalence of periodontitis higher than in the general population.	
PI; BoP; PPD; CAL; ABH		Cases Number of subjective periodontal disease indicators (mean): Baseline: 2.0 18-month PH: 2.4 Change baseline-18 months for clinical parameters (mean ± SE) PI: 0.24 ± 0.37; BoP: 0.02 ± 0.36; PPD: 0.35 ± 0.28 mm; CAL: 0.45 ± 0.38 mm; ABH: 0.32 ± 0.36 mm Controls Number of subjective periodontal disease indicators (mean): Baseline: 1.6 18-month PH: 1.1 Change baseline-18 months for clinical parameters (mean ± SE) PI: 0.36 ± 0.14; BoP: 0.14 ± 0.13; PPD: 0.01 ± 0.22 mm; CAL: 0.03 ± 0.22 mm; ABH: 0.19 ± 0.22 mm			Women taking AI had a significantly worse mean subjective periodontal health score than controls, which tended to worsen during the first 18 months of AI use. Statistically significant greater PPD increase, CAL loss and ABH loss were observed in the case than in the control group, while BoP increased more in the control group. The use of bisphosphonate, vitamin D, and calcium usage reduced ABH loss only in the case group.	
PI; BoP; PPD; AL; REC; ABH; perceived oral health (Likert scale)		Cases Moderate periodontitis: 48.3%; Severe periodontitis: 31% Periodontal parameters (mean ± SD) N° sites with PI: 55.4 ± 3.4 N° sites with BoP: 27.8 ± 23.4 PPD: 2.0 ± 0.27 mm AL: 1.5 ± 0.75 mm Worst site AL: 1.5 ± 0.75 mm REC: 0.36 ± 0.67 mm ABH: 2.65 ± 0.63 mm Perceived dental health: 3.14 ± 0.18 Perceived gum health: 2.97 ± 1.29 Importance of dental health: 4.72 ± 0.75 Controls Moderate periodontitis: 58.3%; Severe periodontitis: 6.9% Periodontal parameters (mean ± SD) N° sites with PI: 16.3 ± 6.6 N° sites with BoP: 16.7 ± 12.3 PPD: 2.0 ± 0.29 mm AL: 1.4 ± 0.39 mm Worst site AL: 1.5 ± 0.75 mm REC: 0.28 ± 0.44 mm ABH: 2.69 ± 0.46 mm Perceived dental health: 3.69 ± 0.96 Perceived gum health: 3.34 ± 1.04 Importance of dental health: 4.97 ± 0.18			Compared with controls, cases had a higher prevalence of severe periodontitis, more sites with BoP, and greater dental plaque, and they tended to rate their oral health lower. In adjusted linear regression (accounting for income, tobacco use, and prior radiation or chemotherapy), AI use was associated with attachment loss exceeding 2 mm (95% CI: 0.46–3.92).	
*Tamoxifen*						
TL		Cases Overall TL (mean ± SD): 2.04 ± 1.58 ≤1 year of drug use (mean ± SD): 1.63 ± 1.77 >1 year of drug use (mean ± SD): 2.32 ± 1.37 Controls TL (mean ± SD): 1.80 ± 1.51			Women who used tamoxifen for more than one year presented greater TL compared to controls as well as those who consumed tamoxifen and did not receive previous chemotherapy or radiotherapy.	
TL based on the M component of DMFT		Cases TL (mean ± SD): 10.45 ± 8.77 Controls TL (mean ± SD): 13.99 ± 8.78			In the adjusted model, the odds of having more than 12 missing teeth were 2.75 times higher among women who used tamoxifen for over one year compared with those treated for less than one year.	
**First Author Year, Country**	**Focused Question**	**Study Design, Study Setting, Time Period, Source of Funding**	**Sample Size (n)**	**Test Group (n)**	**Control Group (n)**	**Definition of Periodontal Disease**
*Studies on breast cancer*						
*Tamoxifen and Aromatase Inhibitors*						
Sun et al., 2025, China [45]	FQ2	Retrospective cohort, Taiwan National Health Registry, January 2010 to December 2019, no external funding	16.492	Women with BC (*n* = 8246), treated with tamoxifen (*n* = 42,746), anastrozole (*n* = 5524), exemestane (*n* = 5705) and letrozole (*n* = 35,654), mean age 55.1 ± 12.3 yrs	Women without BC (*n* = 8246), matched 1:1 in terms of age, income, comorbidities, and urbanization level, mean age 55.6 ± 14.7 yrs	Periodontitis diagnosis based on the International Classification of Diseases, Ninth and Tenth Revision, Clinical Modification (ICD-9-CM and ICD-10-CM)
Park et al., 2023, USA [46]	FQ1	Cross-sectional, NHANES dataset, January 2009 to March 2020, no external funding	423	Adult women with BC undergoing adjuvant endocrine therapy (*n* = 94), of whom 30 assuming tamoxifen, 33 anastrozole, 18 letrozole and 13 exemestane, 54.8% under 65 yrs, 16.6% smokers, 75.8% White, 10.7% Black, 13.5% Other	Adult women with BC not undergoing adjuvant endocrine therapy (*n* = 329), 38% under 65 yrs of age (10.7% smokers), 81.8% White, 6.7% Black, 11.5% Other	NR
Ustaoğlu et al., 2021, Turkey [47]	FQ1 FQ2	Cross-sectional, Department of Medical Oncology Bolu Abant Izzet Baysal University, April 2009 to September 2013, no external funding	155	Women with early-stage BC (Stage I to IIIA) treated with at least one course of endocrine therapy (*n* = 103), of whom:	Systemically healthy women (*n* = 52), mean age 48.33 ± 10.08 yrs, 38 of them in the post-menopausal period.	NR
				- 51 received tamoxifen (mean age 49.70 ± 10.22 yrs) - 52 received AI (mean age 60.04 ± 11.60 yrs).		
				All patients of Al group and 44 of the tamoxifen group in the post-menopausal period.		
de Sire et al., 2021, Italy [48]	FQ1	Cross-sectional, University Hospital “Maggiore della Carità” Novara, January to June 2020, no external funding	122	Post-menopausal women with invasive BC treated with surgery at least 12 months earlier, taking tamoxifen (*n* = 48) or AI (*n* = 74) therapy, mean age 55.6 ± 10.4 yrs, 18% smokers		Periodontal Screening and Recording Index (PSR)
Taichman et al., 2018, USA [49]	FQ1 FFQ2	Cross-sectional, University of Michigan Hospital Michigan, June 2014 to June 2015, no external funding	181	Post-menopausal women with early-stage BC (*n* = 140) on adjuvant treatment for at least 3 months and having > 20 natural teeth, 28 women assuming chemotherapy (mean age 58.0 ± 9.9 yrs, smokers 4%), 52 tamoxifen (mean age 56.3 ± 8.3 yrs, smokers 2%) and 60 AI (mean age 62.5 ± 7.7 yrs, smokers 2%)	Post-menopausal women without BC (*n* = 41), having > 20 natural teeth, mean age 66.0 ± 9.4 yrs, smokers 12%	Subjective perception of teeth and gum health
*Studies on prostate cancer*						
Famili et al., 2007, USA [50]	FQ1	Cross-sectional, University Pittsburgh, no external funding	68	Men with nonmetastatic PC treated with ADT (*n* = 41), mean age 70.5 yrs, 7.3% smokers, 7.3% Black and 92.7% White	Men with nonmetastatic PC not on ADT (*n* = 27), mean age 68.5 yrs, 3.7% smokers, 11.1% Black and 88.9% White	Case of periodontitis defined as at least one site with AL of ≥3 mm
* **Tamoxifen and Aromatase Inhibitors** *						
Risk of developing periodontitis over a mean follow-up time of 6.15 ± 3.01 years		2679 BC women developed periodontitis of whom:			The risk of periodontitis was significantly lower in women who received hormone therapy compared with those who did not.	
		- 1368 assumed tamoxifen (HR = 0.86, 95% IC = 0.79–0.93) - 91 anastrozole (HR = 0.47, 95% IC = 0.38–0.58) - 75 exemestane (HR = 0.38, 95% IC = 0.30–0.48) - 585 letrozole (HR = 0.35, 95% IC = 0.32–0.39)				
Prevalence of gum disease, TL, Number of decayed teeth, Need for immediate dental care		Cases TL (mean ± SE): 12.4 ± 0.51 Coronal cavities (mean ± SE): 1.97± 0.17 Decayed teeth (%): 27.5 Gum disease (%): 27.2 Recommended for imminent dental care (%): 43.4 Controls TL (mean ± SE): 11.7 ± 0.91 Coronal cavities (mean ± SE): 4.44 ± 0.57 Decayed teeth (%): 13.4 Gum disease (%): 13.2 Recommended for imminent dental care (%): 26.1			Endocrine therapy use was associated with increased prevalence of tooth decay and periodontal pathology, and these patients were more often identified as needing prompt dental intervention than those not receiving such therapy.	
PI; BoP; GI; PPD; CAL; N° of decayed teeth; N° of teeth requiring extraction for advanced periodontal involvement, fracture, or extensive carious lesion		Cases (mean ± SE) Tamoxifen users AI users			AI users exhibited the fewest teeth and highest CAL, while PI, GI, and PPD did not differ significantly across groups. PI was lower in patients using AIs for <2 years.	
		PI:	1.74 ± 0.68	1.87 ± 0.54		
		BoP:	58.71 ± 40.20	61.50 ± 39.21		
		GI:	1.60 ± 0.60	1.76 ± 0.33		
		PPD (mm):	2.15 ± 0.70	2.55 ± 1.11		
		CAL (mm):	2.65 ± 0.70	3.34 ± 1.34		
		N° of teeth:	17.00 ± 8.56	15.23 ± 8.23		
		N° of teeth to				
		be extracted:	0.22 ± 0.67	0.23 ± 0.68		
		N° decayed teeth:	0.127 ± 0.105	0.08 ± 0.16		
		Controls (mean ± SE) PI: 1.50 ± 0.53 BoP: 48.40 ± 42.04 GI: 1.52 ± 0.46 PPD (mm): 2.21 ± 0.92 CAL (mm): 2.48 ± 1.39 N° of teeth: 19.17 ± 5.13 N° of teeth to be extracted: 0.63 ± 1.20 N° decayed teeth: 0.22 ± 0.22				
OHI; GBI; PCR; DMFT		Moderate periodontitis: 55.7% Severe periodontitis: 12.2%; insufficient OHI: 53.2%; GBI < 25%: 93.4%; PCR index > 50%: 51.6%; DMFT (mean ± SD): 17.44 ± 6.76			BC women on hormonal therapy showed a high prevalence of mild-to-moderate periodontitis and poor oral care.	
Questionnaire on subjective oral health perception (scale from 1 to 5) and frequency of oral symptoms		Cases (mean ± SD) Chemotherapy Health of teeth: 3.21 ± 0.74 Health of gums: 3.36 ± 0.91 Frequency of teeth sensitive: 1.55 ± 1.01 Frequency of bleeding gums: 1.50 ± 1.00 Frequency of bad breath: 1.67 ± 1.09 Tamoxifen Health of teeth: 3.65 ± 1.08 Health of gums: 3.48 ± 0.98 Frequency of teeth sensitive: 2.25 ± 1.39 Frequency of bleeding gums: 1.78 ± 0.99 Frequency of bad breath: 1.88 ± 0.96 AI Health of teeth: 3.43 ± 0.89 Health of gums: 3.40 ± 0.78 Frequency of teeth sensitive: 1.93 ± 1.20 Frequency of bleeding gums: 1.60 ± 0.96 Frequency of bad breath: 1.95 ± 1.11 Controls (mean ± SD) Health of teeth: 2.73 ± 0.89 Health of gums: 2.93 ± 0.98 Frequency of teeth sensitive: 2.12 ± 1.14 Frequency of bleeding gums: 1.84 ± 0.94 Frequency of bad breath: 2.02 ± 0.97			Controls had worse perception of teeth and gum health compared to BC women irrespective of the drug regimen, but tamoxifen and AI users reported higher frequency of sensitive teeth than chemotherapy users.	
*Studies on prostate cancer*						
PI; BoP; PPD; AL; REC		Cases Prevalence of periodontitis: 80.95% Frequency of tooth mobility: 14.63% Frequency of BoP: 68.3% Frequency of PPD 3–4 mm: 100% Frequency of REC: 90.24% Frequency of AL: 100% Controls Prevalence of periodontitis: 3.70% Frequency of tooth mobility: 0% Frequency of BoP: 25.9% Frequency of PPD 3–4 mm: 3.70% Frequency of REC: 7.41% Frequency of AL: 7.41%			Men with prostate cancer undergoing ADT were more likely to have periodontal disease than men not on ADT despite having similar oral hygiene habits.	

ABH, alveolar bone height; ADT, androgen deprivation therapy; AI, aromatase inhibitor; AL, attachment loss; BC, breast cancer; BoP, bleeding on probing; CAL, clinical attachment level; CI, confidence interval; DMFT, number of decayed, missing and filled permanent teeth; FQ, focused question; GI, gingival index; HR, hazard ratio; NHANES, National Health and Nutrition Examination Survey; OR, odds ratio; PC, prostate cancer; PPD, probing pocket depth; PI, plaque index; PCR, Plaque control recording; PSR, periodontal screening and recording; REC, gingival recession; SD, standard deviation; SE, standard error; TL, tooth loss; WTCI, Winkel tongue coating index.

## Data Availability

All data used in this study are available in the manuscript or its Appendix A.

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
