# Peer review of "Impact of Endocrine Therapy for Cancer on Periodontal Health: A Systematic Review"

_cancers, 2025, doi:10.3390/cancers17183066_

Round 1

Reviewer 1 Report

Comments and Suggestions for Authors

The review article entitled “Impact of Endocrine Therapy for Cancer on Periodontal Health: A Systematic Review” has been evaluated. This study underscores a significant yet frequently overlooked aspect of survivorship care, emphasizing the necessity for routine periodontal assessments and interdisciplinary collaboration in the management of patients undergoing hormone therapy for cancer. Nonetheless, several areas require improvement:

  1. The introduction should be broadened, particularly with respect to cancer statistics. Incorporating detailed regional data will provide essential context and underscore the significance of the research on both local and global levels. Currently, the author presents limited global and local data towards the end of the introduction; this information should be repositioned to the beginning of the section.
  2. The authors have cited a limited number of references regarding the impact of endocrine therapy for cancer on periodontal health. There are numerous review articles that address this topic in depth. It is advisable for the authors to acknowledge some of these studies while distinguishing the unique contributions of the current research.
  3. It is strongly recommended to include a diagram that illustrates the potential relationships between adjuvant endocrine breast cancer treatments, systemic osteopenia/osteoporosis, and oral bone health. Furthermore, I suggest incorporating additional references that are independent of my colleagues and myself, as they are more relevant to this study.
  4. The manuscript should provide a more comprehensive overview of the statistical methods employed throughout the research. Each statistical test used must be clearly justified, as this will enhance the credibility of the study and help readers understand the data analysis processes.
  5. The conclusion warrants thorough revision to effectively convey the broader implications of the study’s findings. The authors should consider discussing the potential applications of their research, emphasizing the significance of endocrine therapy for cancer on periodontal health, and outlining clear directions for future research that will inform subsequent investigations in this area.
  6. Moreover, the current manuscript demonstrates a similarity index of 27%. This percentage should be reduced to below 20% to comply with publication standards.

Author Response

We thank the Editor and the Reviewers for their valuable feedback. Below, we have addressed their comments and concerns, and we have marked the changes in red color in the manuscript. We trust that these revisions address the reviewers' concerns adequately, and we hope that the manuscript can now become suitable for publication in Cancers.

Reviewer 1:

The review article entitled “Impact of Endocrine Therapy for Cancer on Periodontal Health: A Systematic Review” has been evaluated. This study underscores a significant yet frequently overlooked aspect of survivorship care, emphasizing the necessity for routine periodontal assessments and interdisciplinary collaboration in the management of patients undergoing hormone therapy for cancer. Nonetheless, several areas require improvement.

We would like to thank the Reviewer for his/her thoughtful review and we appreciate the reviewer’s valuable feedback. Your insightful comments helped us to strengthen the quality of our paper. The changes are tracked in the revised paper (in red color). In addiction we revised the the English language and the quality of Figures.

Comment 1: The introduction should be broadened, particularly with respect to cancer statistics. Incorporating detailed regional data will provide essential context and underscore the significance of the research on both local and global levels. Currently, the author presents limited global and local data towards the end of the introduction; this information should be repositioned to the beginning of the section.

Response 1: We thank the reviewer for the interesting suggestion. We have moved the paragraph as recommended and have updated and expanded the text with the most recent data on the incidence and trends of the cancers in question

Comment 2: The authors have cited a limited number of references regarding the impact of endocrine therapy for cancer on periodontal health. There are numerous review articles that address this topic in depth. It is advisable for the authors to acknowledge some of these studies while distinguishing the unique contributions of the current research.

Response 2: We appreciate the Reviewer’s suggestion. We expanded the introduction section in order to better address this aspect. (Lines 54-67)

Comment 3: It is strongly recommended to include a diagram that illustrates the potential relationships between adjuvant endocrine breast cancer treatments, systemic osteopenia/osteoporosis, and oral bone health. Furthermore, I suggest incorporating additional references that are independent of my colleagues and myself, as they are more relevant to this study.

Response 3: We thank the Reviewer for this advice. A diagram illustrating the relationship between endocrine therapy in breast cancer and periodontitis has been now added to the manuscript. (Figure-1).

Comment 4: The manuscript should provide a more comprehensive overview of the statistical methods employed throughout the research. Each statistical test used must be clearly justified, as this will enhance the credibility of the study and help readers understand the data analysis processes.

Response 4: Thank you for your valuable feedback. We have revised the Statistical Analysis section to

provide a more comprehensive and detailed description of the statistical methods employed in the study. (Lines 209-222)

Comment 5: The conclusion warrants thorough revision to effectively convey the broader implications of the study’s findings. The authors should consider discussing the potential applications of their research, emphasizing the significance of endocrine therapy for cancer on periodontal health, and outlining clear directions for future research that will inform subsequent investigations in this area.

Response 5: We thank the Reviewer for this valuable recommendation. In the discussion section we have further clarified the future implication that our work is proposing. (Lines 555-564).

Comment 6: Moreover, the current manuscript demonstrates a similarity index of 27%. This percentage should be reduced to below 20% to comply with publication standards.

Response 6: We thank the reviewer for checking the overlap rate of our review, something we admittedly had not done carefully. Following their observation and the suggestion to revise the English, we have rewritten certain sections of the review. The overlap percentage, excluding the bibliographic entries, is now 18%. We would like to point out that most of the remaining overlap pertains to keywords, definitions, and attributions that we cannot modify, but we are confident that the reviewer will recognize the effort we have made in addressing their comments.

Reviewer 2 Report

Comments and Suggestions for Authors

The authors analyze in this review an important side effect in patients affected by BC and Pc undergoing AI and Tamoxifene therapy: drug-related periodontal disease and tooth loosening.

Even if the reviews show heavy limits as stated by the Authors because of "  Significant heterogeneity was observed among the included studies in terms of design, cancer stage, definitions of Cancers, periodontal disease, methods of periodontal parameter assessment, and adjustment for confounding factors. These inconsistencies limit the comparability of results and hinder the ability to draw definitive conclusions. Moreover, the overall quality of evidence was  low, with a high risk of bias across studies, further restricting the strength of the findings."Nevertheless, the review shows the strength of analyzing by a systematic review a paramount aspect for patients undergoing AI and Tamoxifen endocrine therapy: the periodontal disease, with its high impact on patients' condition and therapy results. Interenstingly two studies reported conflicting data, probably because of periodontal care administered in advance to the patients, highlighting the importance of more studies and periodontal care in these patients. 

Author Response

We thank the Editor and the Reviewers for their valuable feedback. Below, we have addressed their comments and concerns, and we have marked the changes in red color in the manuscript. We trust that these revisions address the reviewers' concerns adequately, and we hope that the manuscript can now become suitable for publication in Cancers.

Reviewer 2:

The authors analyze in this review an important side effect in patients affected by BC and Pc undergoing AI and Tamoxifene therapy: drug-related periodontal disease and tooth loosening.

Comment 1: Even if the reviews show heavy limits as stated by the Authors because of “Significant heterogeneity was observed among the included studies in terms of design, cancer stage, definitions of Cancers, periodontal disease, methods of periodontal parameter assessment, and adjustment for confounding factors. These inconsistencies limit the comparability of results and hinder the ability to draw definitive conclusions. Moreover, the overall quality of evidence was low, with a high risk of bias across studies, further restricting the strength of the findings. "Nevertheless, the review shows the strength of analyzing by a systematic review a paramount aspect for patients undergoing AI and Tamoxifen endocrine therapy: the periodontal disease, with its high impact on patients' condition and therapy results. Interestingly two studies reported conflicting data, probably because of periodontal care administered in advance to the patients, highlighting the importance of more studies and periodontal care in these patients. 

Response 1: We thank the Reviewer for this valuable comment, and we agree that this underexplored field needs further study given the growing number of patients on endocrine therapy and the importance of preserving their oral health during this challenging medical and personal circumstance.

Reviewer 3 Report

Comments and Suggestions for Authors

This paper addresses the important theme of the impact of cancer treatment on oral health, and its clinical significance is substantial. However, particularly due to its detailed examination of periodontal tissues, publication in a dental specialty journal, especially one focusing on periodontology or oral medicine, might be more appropriate for disseminating this information to specialists in this field.

Author Response

We thank the Editor and the Reviewers for their valuable feedback. Below, we have addressed their comments and concerns, and we have marked the changes in red color in the manuscript. We trust that these revisions address the reviewers' concerns adequately, and we hope that the manuscript can now become suitable for publication in Cancers.

Reviewer 3

Comment 1: This paper addresses the important theme of the impact of cancer treatment on oral health, and its clinical significance is substantial. However, particularly due to its detailed examination of periodontal tissues, publication in a dental specialty journal, especially one focusing on periodontology or oral medicine, might be more appropriate for disseminating this information to specialists in this field.

Response 1: We thank the Reviewer for this thoughtful observation and we fully recognize the value of disseminating this information to specialists in oral medicine and periodontology. Yet, we deliberately selected a cancer-oriented journal, as raising awareness among oncologists—the primary clinicians managing oncological patients—is crucial to stimulate a more multidisciplinary care which represents a cornerstone for improving the quality of life of individuals undergoing endocrine therapy.

Round 2

Reviewer 1 Report

Comments and Suggestions for Authors

The authors made significant revisions in response to the reviewers' comments. The manuscript can be acceptable for publication.

Author Response

Reviewer 1 comment

Comment: The authors made significant revisions in response to the reviewers' comments. The manuscript can be acceptable for publication.

Author’s response: We are sincerely grateful to the Reviewer for dedicating time and expertise to provide valuable feedbacks, which has significantly improved our manuscript.